# Functional correlates of cognitive performance and working memory in temporal lobe epilepsy: Insights from task-based and resting-state fMRI

Alfonso Fajardo-Valdez[1☉], Vicente Camacho-Téllez[1,2,3☉], Raúl Rodríguez-Cruces[1,4], María Luisa García-Gomar[5], Erick Humberto Pasaye[1], Luis Concha[1] *

1 Institute of Neurobiology, Universidad Nacional Autónoma de México, Campus Juriquilla, Querétaro, México, 2 Department of Psychiatry, FLENI, Buenos Aires, Argentina, 3 Consejo Nacional de Investigaciones Científicas y Técnicas (CONICET), Buenos Aires, Argentina, 4 Multimodal Imaging and Connectome Analysis Lab, McConnell Brain Imaging Centre, Montreal Neurological Institute, McGill University, Montreal, Québec, Canada, 5 Facultad de Ciencias de la Salud, Universidad Autónoma de Baja California, Tijuana, México

☉ These authors contributed equally to this work.
* lconcha@unam.mx

## Abstract

Temporal lobe epilepsy (TLE) is a common form of medically intractable epilepsy. Although seizures originate in mesial temporal structures, there are widespread abnormalities of gray and white matter beyond the temporal lobes that negatively impact functional networks and cognition. Previous studies have focused either on the global impact on functional networks, or on the functional correlates of specific cognitive abilities. Here, we use a two-pronged approach to evaluate the link between whole-brain functional connectivity (FC) anomalies to overall cognitive performance, and how such abnormal connectivity alters the fronto-parietal brain regions involved in working memory (WMem), a cognitive disability often reported by TLE patients. We evaluated 31 TLE patients and 35 healthy subjects through extensive cognitive testing, resting-state functional magnetic resonance imaging (RS-fMRI), and task-based fMRI using Sternberg's task to evaluate WMem. As a group, TLE patients displayed cognitive abnormalities across different domains, although considerable within-group variability was identified. TLE patients showed disruptions of functional networks between and within the default mode network (DMN) and task-positive networks (TPN) resulting in associations with cognitive performance. Furthermore, during the WMem task, TLE patients showed abnormal activity of fronto-parietal regions that were associated with other forms of memory, and alterations of seed-based connectivity analyses. Our results show that different degrees of abnormal functional brain activity and connectivity are related to the severity of disabilities across cognitive spheres. Differential co-activation patterns between patients and healthy subjects suggest potential compensatory mechanisms to preserve adequate cognitive performance.

**Data Availability Statement:** All data, including raw MRI images (T1-weighted, resting-state fMRI and task-based fMRI), and cognitive scores, are freely-available at OpenNeuro (data set ds004469, https://doi.org/10.18112/openneuro.ds004469.v1.1.3). Additionally, code for statistical models and interactive visualizations of voxelwise results are at https://github.com/alffajardo/TLE2023_fMRI

**Funding:** This work was supported by CONACYT (181508, 1782, FC218-2023); and UNAM-DGAPA (IB201712, IG200117, IN204720, IN213423). Raúl Rodríguez-Cruces and Alfonso Fajardo received fellowships from Conacyt (329866 and 478686). The funders had no role in study design, data collection and analysis, decision to publish, or preparation of the manuscript. There was no additional external funding received for this study.

**Competing interests:** The authors have declared that no competing interests exist.

## Introduction

Temporal lobe epilepsy (TLE) is the most common type of focal epilepsy [1]. TLE seizures are often refractory to antiepileptic drugs (AEDs), and due to this, TLE usually entails large detrimental effects on the quality of life of patients, affecting their physical and mental health, with a wide spectrum of cognitive deficits [2, 3]. Long-term memory deficits are one of the most frequent concerns in these patients [4, 5], with self-reported memory being an important predictor of quality of life that correlates with objective performance on memory tests [2]. Notably, cognitive deficits are already present in newly-diagnosed epilepsy in children [3], with potential for further deterioration due to long-term pharmacological treatment [6]. A large body of research has reported that TLE patients typically undergo difficulties in their ability to recall, often displaying important short- and long-term memory deficits [7]. It is also well known that the hippocampus and medial temporal lobes play critical roles in declarative memories [8, 9]. Indeed, patients who exhibit neuronal loss, volume reduction, and gliosis in mesial temporal structures (mesiotemporal sclerosis, MTS) also display the most severe deficits in short- and long-term memory [7, 10]. Notably, structural and functional abnormalities are not restricted to the temporal lobe, with a large body of evidence showing alterations widespread throughout the brain, comprising neocortical thinning, atrophy of subcortical structures, and abnormalities of white matter [11–13]. Such widespread alterations are linked to deficits in cognitive domains not necessarily or exclusively linked to temporal lobe function, such as executive functions, attention, and working memory [14–19].

Resting-state functional connectivity (RSFC) has been used to show abnormalities of the default mode network (DMN), attention, and reward/emotion networks that may partially explain cognitive deficits across domains [20, 21]. Cognitive abilities of healthy participants are correlated with the adequate interplay of the DMN and task-positive networks (TPN) [22, 23]. Nonetheless, it is unknown how the altered networks of TLE patients modulate the interaction between large-scale functional connectivity (FC) and specific cognitive abilities.

The concept of working memory (WMem) was first described by Miller in 1960 and refers to a type of short-term memory which has a crucial cognitive function that supports ongoing and upcoming behaviors, allowing storage of information across delay periods [24]. A basic feature of WMem is the maintenance of information in the short term in the absence of any sensory input [25]. This means that once the information is presented, the individual is able to retain and manipulate it without the stimulus remaining present. WMem is a basic component of complex cognitive functions such as learning and reasoning [26]. Crucially, WMem does not appear to involve temporal lobe structures, but rather maintenance of information in WMem is carried out within a fronto-parietal network [25]. WMem appears to be further impaired in patients with left TLE with hippocampal sclerosis, early onset of the disorder, and high frequency of seizures [27]. Functional Magnetic Resonance Imaging (fMRI) has shown bilateral activation of the frontal and parietal lobes using paradigms to evaluate WMem (e.g., N-back) [28], and such activity is impaired in TLE patients [18, 29]. In healthy subjects, the hippocampus is normally bilaterally deactivated during WMem tasks, and the level of such deactivation is dependent on memory load. Interestingly, this pattern is abnormal in patients with TLE [30]. While group studies have shown that TLE patients have WMem deficits [29–32], the impact of WMem patterns of network co-activity on overall cognitive performance of TLE patients is unknown.

In this work, we explored the association between brain function and cognitive performance in TLE patients using two complementary approaches. First, we explore the link between cognitive scores across domains with the activity of resting-state functional networks. Next, we use task-based fMRI to specifically investigate WMem abilities and fronto-parietal

network co-activity. Finally, we investigate the interplay between WMem networks resulting from task-fMRI and those derived from resting-state fMRI, and how both impact on overall cognitive abilities.

## Materials and methods

### Participants

We recruited a sample of 31 patients with TLE, previously reported in [17] (age 10 ± 11 years old, 22 women). Patients were diagnosed and followed at out-patient epilepsy clinics at Hospital General de México in Mexico City, and at Hospital Central "Dr. Ignacio Morones Prieto" in San Luis Potosí, México. The referring neurologists diagnosed the patients according to ILAE criteria, relying on clinical information, surface EEG, and conventional neuroimaging. Based on electroclinical and neuroimaging data, the laterality of the epileptogenic focus was determined, resulting in 19 left-TLE, and 12 right-TLE. Seventeen patients had imaging evidence of mesial temporal sclerosis. None of the patients were at the time using barbiturates, benzodiazepines, or topiramate—all known to affect cognitive functions. The control group consisted of 35 healthy subjects matched for age and education (age 33 ± 12 years old; 17 women) without any history of neurologic or psychiatric disease. All participants were right-handed. Specific sample sizes differed between sub-analyses, as indicated below, summarized in S1 Fig, and annotated in the accompanying dataset. The study was approved by the Ethics Committee of the Institute of Neurobiology and written informed consent was obtained from all subjects in accordance with the Council for International Organizations of Medical Sciences (CIOMS) International Ethical Guidelines for Health-Related Research Involving Human Subjects.

### Neuropsychological assessment

Participants underwent extensive cognitive assessment at Hospital General de México and Centro Estatal de Salud Mental, Querétaro, México, which was conducted by experienced neuropsychologists. Two scales were used: Wechsler's Adult Intelligence Scale (WAIS-IV) and Weschsler's Memory Scale (WMS-IV). Scores of these tests were reported as percentiles adjusted for age and education level based on the Mexican population. Derived from these two scales, we used nine indices as measurements of performance on specific cognitive domains: verbal comprehension (VCI), perceptual reasoning (PR), processing speed (PSI), working memory (WMI), visual memory (VMI), visual working memory (VWMI), immediate memory (IMI), delayed memory (DMI), and auditory memory (AMI).

### Magnetic resonance imaging

We acquired resting-state (RS) and task-fMRI sequences. Studies were performed on a 3 T Philips Achieva TX scanner (Best, The Netherlands) at the National Laboratory for MRI, located at Instituto de Neurobiología, UNAM Campus Juriquilla, Querétaro, Mexico. For the acquisition of RS-fMRI, participants were instructed to close their eyes, stay still and relaxed, and to stay awake during the scanning session. Gradient-echo echo-planar T2*-weighted images (TR/TE = 2000/30 ms) were acquired covering the whole brain with 34 slices, with a voxel resolution of 2×2×3 $mm^3$ (200 volumes; 6 minutes 40 seconds). Similarly, task-fMRI (190 volumes; 6 minutes 20 seconds) was acquired with the same scan parameters while subjects performed the Sternberg memory task (described below). For registration purposes, we also acquired high-resolution T1-weighted images with 1×1×1$mm^3$ voxel size using a 3D magnetization-prepared gradient echo sequence (TR/TE = 8.1/3.7 ms; flip angle = 8˚). Additionally, other structural and DWI sequences were acquired and described elsewhere [17, 33]. We

conducted a quality assessment on fMRI images, and the exclusion criteria for subjects were independent for each modality. Therefore, the number of included subjects differs slightly between RS-fMRI and task-fMRI analyses and is reported in each section (these subsamples did not differ on sex, age, and education levels).

### Sternberg's memory task

We used a modified version of Sternberg's memory task [34] which was introduced to participants prior to the imaging session, making sure they understood and complied with the task. Participants laid on the scanner and viewed a series of slides through a mirror affixed to the MRI head coil. Each trial consisted of three slides (Fig 1), which were presented using E-Prime version 2 (Psychology software tools, Sharpsburg, PA): The first slide showed a list of either 2, 4, or 6 digits which participants were instructed to memorize (encoding phase, 4 s duration); the second slide showed only three dots that indicated that the previous list of digits must be retained in memory (retention phase, 10 s); the last slide showed a single digit between question marks, and the subject was asked to respond whether that specific digit was included in the list of digits seen in the first slide (retrieval phase, 4 s). A total of 12 trials with different memory loads and digits in each list were evaluated per participant, with an inter-trial period of 18 seconds. Participants provided their response (yes/no) through MRI-compatible triggers (Nordic NeuroLab, Bergen, Norway) by pressing a button with either index finger; the side indicating yes or no was counterbalanced between participants. Participants were instructed to provide their responses as quickly as possible. The number of correct responses and the reaction times were analyzed as measures of performance in the task, through mixed 2x3 analyses of variance (ANOVA) considering group (TLE or Control) as the between-subjects factor and memory load (2, 4, or 6 digits) as the within-subjects factor. Statistical significance was considered at a p-value < 0.05.

### Resting-state fMRI preprocessing

RS-fMRI datasets were preprocessed with the Configurable Pipeline for the Analysis of Connectomes (C-PAC v.1.1.8) [35], set with the default parameters with minimal adjustments

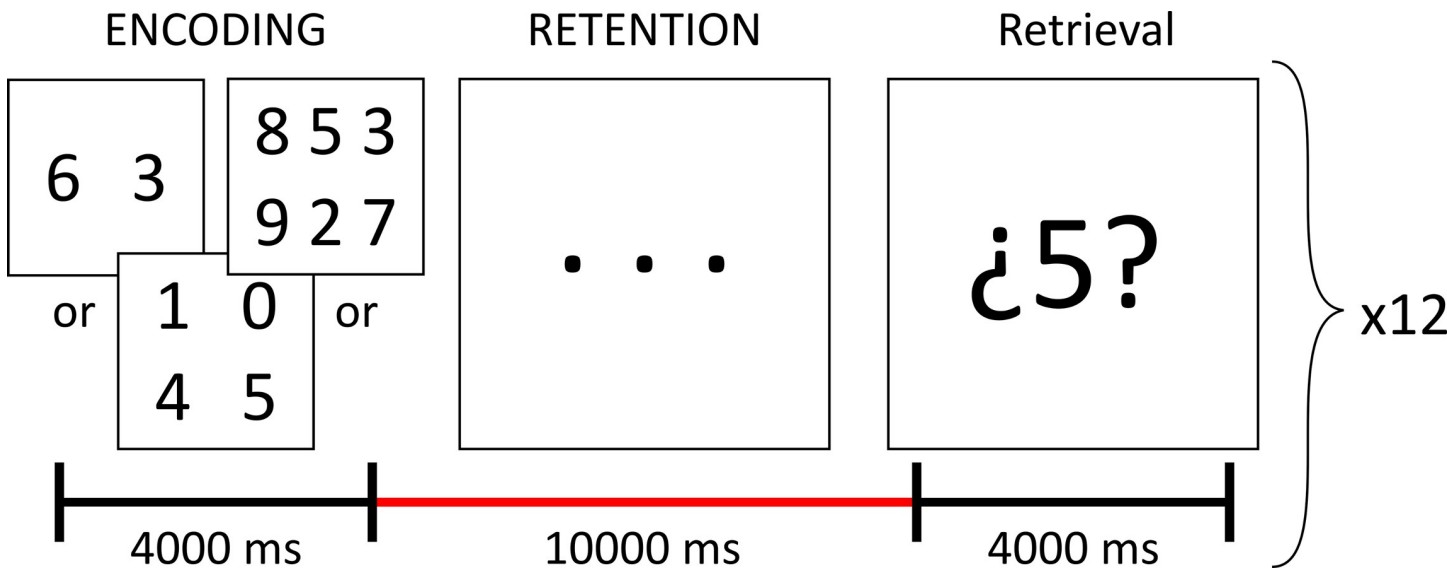

**Fig 1. Sternberg memory task for evaluation of WMem.**

based on [36, 37]. The preprocessing pipeline included the following procedures: slice-timing correction for an interleaved acquisition, motion correction, skull and non-brain tissue removal, alignment of functional timeseries to the anatomical T1-weighted volume, nuisance artifact removal (nuisance regression), spatial normalization, and spatial smoothing. The matrix of nuisance regressors included linear and quadratic trends, white matter (WM) mean intensity, cerebrospinal fluid (CSF) mean intensity, the top 6 principal components derived from WM and CSF signal (aCompcor), motion parameters and their derivatives [38], and 0.01–0.08 Hz bandpass filtering. Additionally, framewise displacement (FD) was calculated from rigid motion parameters, and timepoints whose FD value was greater than 0.25 were marked as spikes and added to the nuisance regressors matrix (volume censoring or soft scrubbing) to prevent spurious results in posterior analyses. Following nuisance regression, residuals were spatially normalized and warped to the Montreal Neurological Institute 152 atlas (MNI152, voxel resolution of 2x2x2 mm$^3$), and data was spatially smoothed through the application of a gaussian kernel of 6 mm FWHM. After visual quality assessment of preprocessed data, datasets with excessive head motion, defined as remaining with less than 120 non censored timepoints (i.e., less than 4 minutes of usable signal) were discarded from the study.

### Definition of DMN and TPN regions of interest

Regions of interest (ROIs) of DMN and TPN were obtained from neurosynth.org [39]. Briefly, we selected the MNI152 seed voxel coordinates, x = -8, y = -56, z = 26, corresponding to the posterior cingulate cortex (PCC) [40]. Seed coordinates were then used in the neurosynth's locations tool to generate a seed-based Pearson coefficient correlation whole-brain map based on a sample of 1000 RS-fMRI datasets (see https://neurosynth.org/locations/ for more details). Next, the seed map was thresholded, keeping only r values lower than -0.25 or greater than 0.25, and the remaining voxels were grouped into clusters, resulting in 21 clusters, 8 corresponding to DMN regions and the remaining 13 corresponding to task-positive regions (DMN anticorrelations). For the purpose of analyzing regions with an equivalent number of voxels, these clusters were used as a template to generate 4mm radius spherical ROIs centered at each cluster's local maxima (see Table 1).

### Functional connectivity and network analysis

Out of the 53 preprocessed RS-fMRI subjects, 2 were excluded due to severe signal dropout in the temporal lobes, and 5 were discarded due to an insufficient number of volumes after data scrubbing. Consequently, data from 46 subjects remained usable. Among them, 25 were controls (18 women), and 21 were patients (14 women, 12 with left-TLE, 9 with right-TLE). To enhance the sensitivity of our analyses, given the substantial exclusion of TLE datasets, we formed a homogeneous TLE group by flipping the brains (inverting the x-axis of images) of right TLE patients. This approach, widely employed in epilepsy studies [41, 42, 51], increases sensitivity at the expense of reduced specificity. Therefore, the results of RS-fMRI analyses are referenced to the location of the epileptogenic focus (ipsilateral/contralateral) rather than to the hemispheric (left/right) location of brain regions. The ROIs mean timeseries were extracted from each preprocessed RS-fMRI dataset, and we obtained 21x21 FC matrices through the calculation of Pearson's correlation coefficients between timeseries of each pair of ROIs. We then carried out statistical comparisons of FC between patients and controls, through node-based and network-based approaches. For the node-based approach, matrices were linearized and FC from each pair of nodes was compared between groups, yielding computation of multiple two-tailed Student's t-tests for independent samples, followed by adjustment of p-values with false discovery rate (FDR). Group differences were also assessed in the

**Table 1. Regions of interest.** Spatial coordinates of local maxima in MNI space (mm).

| Region of interest (ROI) | Abbreviation | X | Y | Z |
|---|---|---|---|---|
| Contralateral Hippocampus | C.HIPP | 26 | -16 | -16 |
| Ipsilateral Hippocampus | I.HIPP | -24 | -18 | -16 |
| Contralateral Middle Temporal Gyrus | C.MTG | 62 | -4 | -16 |
| Ipsilateral Middle Temporal Gyrus | I.MTG | -62 | -8 | -14 |
| Contralateral Posterior Parietal Lobe | C.PPL | 52 | -60 | 32 |
| Ipsilateral Posterior Parietal Lobe | I.PPL | -50 | -66 | 34 |
| Posterior Cingulate Cortex | PCC | -8 | -56 | 26 |
| VentroMedial Prefrontal Cortex | VmPFC | 0 | 54 | -6 |
| Ipsilateral Middle Temporal Gyrus (Temporal-Occipital part) | I.MTG-to | -56 | -62 | 0 |
| Contralateral Middle Temporal Gyrus (Temporal-Occipital part) | C.MTG-to | 60 | -54 | -4 |
| Ipsilateral Precentral Gyrus | I.PrG | -28 | -6 | 52 |
| Ipsilateral Parieto-Occipital Sulcus | I.POS | -18 | -70 | 50 |
| Contralateral Parieto-Occipital Sulcus | C.POS | 18 | -70 | 54 |
| Paracingulate Gyrus | PCG | 4 | 12 | 52 |
| Ipsilateral Insula | I.INS | -32 | 16 | 8 |
| Contralateral Insula | C.INS | 34 | 18 | 6 |
| Ipsilateral Frontal Pole | I.FP | -38 | 42 | 34 |
| Contralateral Frontal Pole | C.FP | 38 | 46 | 30 |
| Contralateral Precuneus | C.PrC | 14 | -34 | 44 |
| Ipsilateral Supramarginal Gyrus | I.SMG | -62 | -36 | 38 |
| Contralateral Supramarginal Gyrus | C.SMG | 62 | -32 | 42 |

presence of covariates 1) age at diagnosis, 2) sex, 3) participant's age, 4) presence/absence of MTS, and 5) Normalized Ipsilateral/contralateral hippocampal volumes (S3 Fig).

A network-based approach was conducted using the Network-Based statistic (NBS) toolbox [43] on MATLAB v.2018a. Similarly to the cluster correction in voxelwise analyses, NBS focuses on the computation of cluster statistics rather than on independent node statistics, and thus, it yields a higher sensitivity and statistical power at the cost of reducing the specificity of statistical comparisons. To conduct NBS analysis, first, an initial cluster threshold value is chosen to identify significant connections between pairs of nodes. Next, connections that surpass such thresholds are used to reveal the distinct components of the network. Lastly, statistical comparisons between samples are carried out, and a p-value is assigned to the components through a permutation test. In this study, we tested between-groups hypotheses bidirectionally (Controls > Patients; Controls < Patients) with an initial threshold set at t-test > 2.1 and subsequent permutations test set at n = 5000. In both the network-based and the node-based approaches, results were deemed significant at a p-value < 0.05.

## Association between resting-state functional connectivity and neuropsychological scores

We aimed to examine whether FC of the Default Mode Network (DMN) and TPN correlated with performance in various cognitive domains in both the whole sample and groups. Accordingly, to explore global associations, we fitted simple linear regressions (GLM) to predict each cognitive score from FC of node pairs ($CS = \beta_0 + \beta_1 FC_{ij} + \beta_2 Group + \varepsilon rror$), and to explore group-specific associations we fitted analysis of covariance (ANCOVA) models predicting each cognitive score as an interaction between FC of node pairs and group ($CS = \beta_0 + \beta_1 FC_{ij} + \beta_2 Group + \beta_3 FC_{ij}*Group + \varepsilon rror$). When reporting results of GLM and ANCOVA analyses, we

focused the significance of the $\beta_1 FC_{ij}$ and $\beta_3 FC_{ij}*Group$ terms respectively. Since we built an independent model for each functional connection, we controlled for false positives by calculating an adjusted p-value threshold using Bonferroni's method ($\alpha/n$, with $\alpha = 0.05$ and $n = 9$ cognitive indexes, resulting in $p_{corr} < 0.0055$. Additionally we evaluated the same GLM/ANCOVA models adjusting for covariates: 1) age at seizure onset, 2) sex, 3) participant's age, 4) presence/absence of MTS, and 5) normalized Ipsilateral/contralateral hippocampal volumes (see S3 Fig).

## General linear model of fMRI during Sternberg's memory task

The analyzed sub-sample consisted of 62 subjects, from which 33 were controls (26 women) and 29 were TLE patients. Images were preprocessed with fMRIprep v.1.5.5 [44]. The preprocessing pipeline for task-fMRI sequences was similar to the RS-fMRI pipeline except that we applied high-pass filtering instead of bandpass filtering, and volume censoring and nuisance regression steps were not carried out during preprocessing, and instead a matrix of nuisance regressors (as described in fmriprep documentation) was calculated and stored for its later use first-level analyses. As for the RS-FC analysis, right-TLE images were swapped on the x-axis. To evaluate the WMem network we used the FEAT tool in FSL (version 6.00, FSL tools, FMRIB, Oxford) [45], and carried out a general linear model (GLM) for a block design on each subject's image series. First-level analyses were carried out by modeling the BOLD signal time series as a linear combination of six main regressors: one regressor for each of the three stages of the task (encoding, retention, retrieval) and one for each of the three levels of difficulty (recalling of 2, 4, or 6 digits), convolved with a gamma hemodynamic response function. Additionally, we generated regressors of no interest (covariates) for: trials in which participants answered incorrectly, rigid motion parameters and the top 10 aCompcor regressors obtained during preprocessing. First-level contrast parameter estimates (Copes) were used for higher level analyses to investigate fMRI activation at the group-level during WMem processes. Only the retention phase of the WMem paradigm was analyzed, since it is difficult to dissociate attention from motion initiation processes secondary to encoding and retrieval phases, respectively. Single-group average fMRI activation was assessed through a mixed effects (FLAME 1) model. Correction for multiple comparisons was carried out through random field theory (cluster forming threshold: z>3.1), and regions were deemed significant if $p_{cluster} < 0.05$.

## Association between BOLD activity during retention phase and cognitive abilities

From identified clusters involved in the retention phase, we generated spherical regions of interest (ROI; 4mm radius) centered at the local maxima (Table 1). Considering previous reports of TLE patients displaying aberrant hippocampal activity during working memory [18, 46, 47], we also included two spherical hippocampal ROIs derived from the DMN-TPN networks resulting from RS analyses. Mean BOLD percentage change during the retention phase relative to basal activity was obtained per ROI and was analyzed and compared between groups through Student's t or Mann-Whitney U tests. Additionally, we examined whether such BOLD activity was correlated with neuropsychological performance by fitting GLM or ANCOVA models as described above for FC analysis. Each cognitive score was predicted as a function of BOLD activity in GLMs and as an interaction between BOLD activity and group in ANCOVA per ROI, with statistical significance considered at $p_{FDR} < 0.05$.

Finally, we explored whether regions involved in the retention phase of Sternberg's memory task were functionally altered during rest. For this purpose each ROI was used to generate a whole-brain FC map through computation of Pearson's correlation coefficient of the ROI's time series and the time series of every other voxel. FMRIB-FSL Randomize (version v.2.9)

was used with 5000 permutations to determine p-value followed by threshold-free cluster enhancement to detect significant clusters.

## Data availability

All data, including raw MRI images (T1-weighted, resting-state fMRI and task-based fMRI), and cognitive scores, are freely-available at Openneuro (data set ds004469, https://doi.org/10.18112/openneuro.ds004469.v1.1.3). Additionally, code for statistical models and interactive visualizations of voxelwise results are at https://github.com/alffajardo/TLE2023_fMRI.

## Results

### Neuropsychological tests

Analysis of cognitive indices derived from WAIS-IV and WMS-IV batteries showed that control subjects obtained average scores according to the population norm. In contrast, scores of TLE patients reflected mild to moderate cognitive impairment, with significantly lower scores in all the cognitive domains as compared with healthy subjects (Fig 2). After Bonferroni correction, all differences remained significant with the exception of VWMI. Scores of TLE

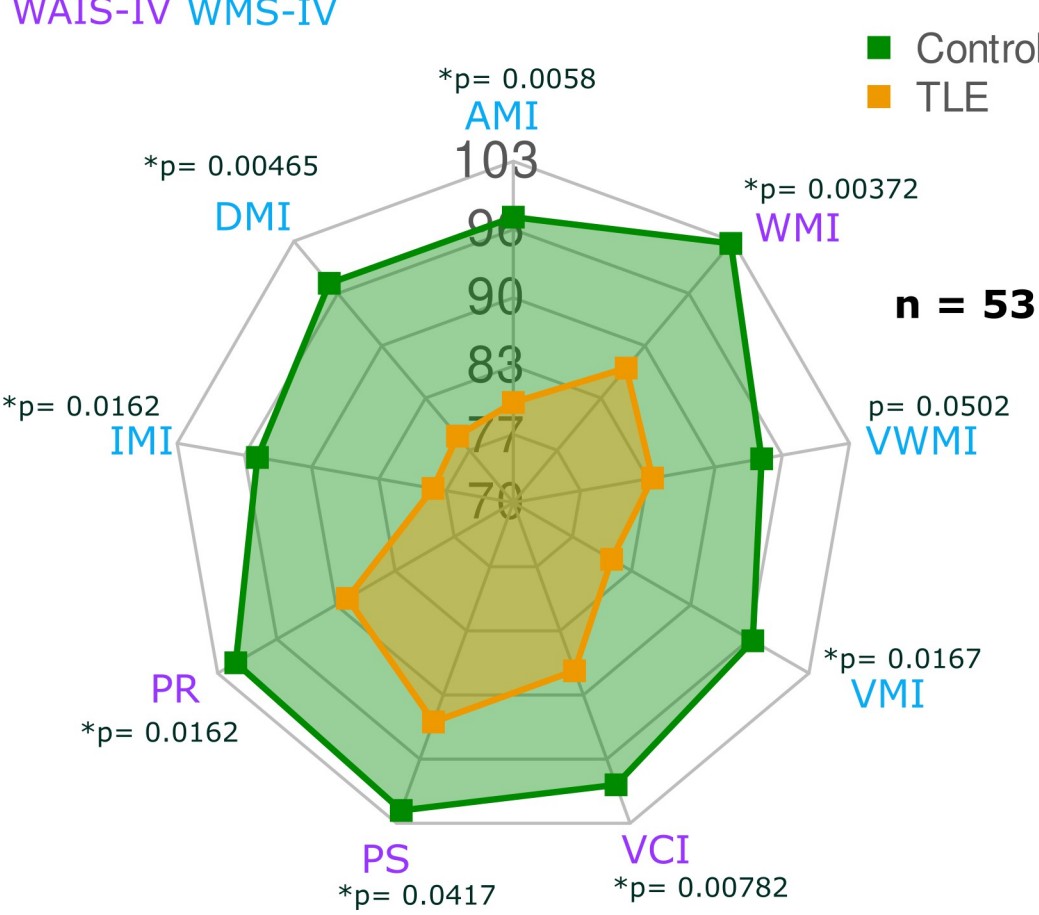

**Fig 2. Cognitive indexes.** Results are represented by group means. Significantly different indexes between groups are labeled with an asterisk (*) for Bonferroni corrected p < 0.05. Nearly identical results are observed when analyzing the subgroups used for each of the imaging analyses (S2 Fig).

patients were lowest in indexes related to memory recall and/or retention of information. As sample sizes varied slightly between imaging analyses, we explored the cognitive profiles in each sub-sample of participants, showing similar between-group differences (S2 Fig).

## Network analysis

Analysis carried out with NBS revealed the presence of three components (sub-networks) that were significantly different between healthy participants and TLE patients. As shown in Fig 3, Component 1 corresponds to a sub-network of the DMN which included both hippocampi, both middle temporal gyri, ventromedial prefrontal cortex and posterior cingulate cortex. Compared to healthy participants, TLE patients displayed a decrease of FC in this component. Similarly, TLE patients also showed a decrease in FC in Component 2 (Fig 3A and 3C), a sub-network of TPN, that involves nodes from the salience network (anterior insular cortices and paracingulate gyrus) and the contralateral supramarginal gyrus. On the other hand, while the control group showed the typical anticorrelation between DMN and TPN, patients with TLE did not show such divergent activity amongst these two networks, involving structures also seen in Component 1, such as the ipsilateral hippocampus and middle temporal gyrus. In addition to the NBS approach, we searched for alterations in FC at individual connections through t-tests between groups (Fig 3E). After FDR correction, C.HIPP-I.HIPP and PCG-I.INS were significantly lower in TLE patients ($p_{corr} < 5{\times}10^{-6}$ for both).

We repeated this analysis using clinical information as covariates, including age at seizure onset, sex, and hippocampal volume. Overall, edge-wise results were similar at the uncorrected p value (S3 Fig), Notably, nine edges consistently showed between-group differences in all the explored models, involving both hippocampi, middle temporal gyri, and contralateral insula and supramarginal gyrus. However, these models did not show statistically-significant differences after correction with FDR and NBS.

## Associations between functional connectivity and cognitive scores

Fig 4 shows significant associations between FC edges and cognitive scores after correction for multiple comparisons. As depicted in panel A, cognitive scores were significantly predicted from FC of edge pairs regardless of subject's groups. Roughly, we observed that intra-network connectivity was positively correlated with cognitive performance while inter-network connectivity (DMN-TPN anticorrelations) was negatively correlated with cognitive performance (See also S4C Fig). Interestingly, most of the node pairs involved in such associations, also differed between groups at the network level as shown in Fig 3.

We also sought for group-specific associations through ANCOVA models. TLE patients showed several significant correlations between FC and cognitive scores, while healthy subjects did not. These associations involved both hippocampi, middle temporal lobe regions, insular cortices, and Ipsilateral precentral gyrus (Figs 4B and S4D).

## Sternberg's memory task

**Behavioral paradigm.** Control group and TLE patients did not differ in the number of correct responses (F(1,67) = 2.23, p = 0.139). Across both groups, the number of correct responses during Sternberg's task correlated with working memory index (r = 0.47, p = 0.0008). A significant effect of cognitive load (number of digits retained in memory) was found (F(2,134) = 7.991, p < 0.001). Post-hoc Bonferroni-corrected tests revealed that both groups performed significantly better when they had to memorize 2 vs 6 digits (t(32) = 2.97, p < 0.05; t(35) = 3.01, p < 0.05). In addition, the control group also performed significantly better when they had to memorize 2 vs 4 digits (t(32) = 2.97, p < 0.05). No interaction between

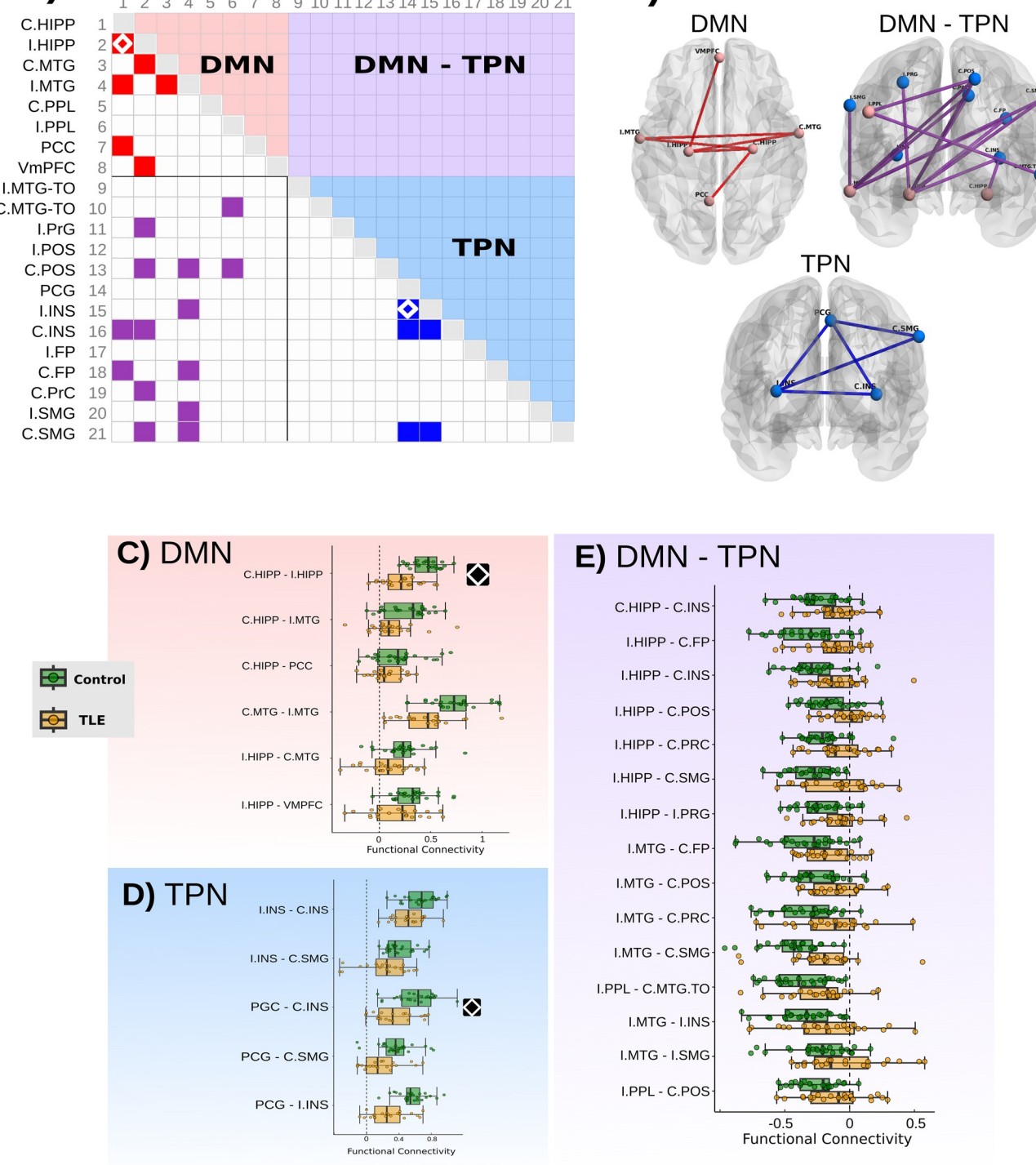

**Fig 3. Functional connectivity differences between controls and TLE patients.** A): Adjacency matrix showing between-group pairwise connectivity differences. Full-colored cells depict group differences at an uncorrected p-value<0.05, while diamond-filled cells indicate significance after FDR correction. color-coded is used to depict the three components identified with NBS analysis (default-mode network: DMN; task-positive networks: TPN; and their interaction: DMN-TPN). Their corresponding spatial configuration is shown in B. C-E: Boxplots show group-wise values at the edge level for each component. Diamonds in A, and C-E represent significant between-group differences in pairwise node-node comparisons (FDR $p_{corr}<0.05$). Refer to Table 1 for ROI abbreviations. Brain slices and surfaces in this and Figs 4–6 correspond to the MNI 152 atlas.

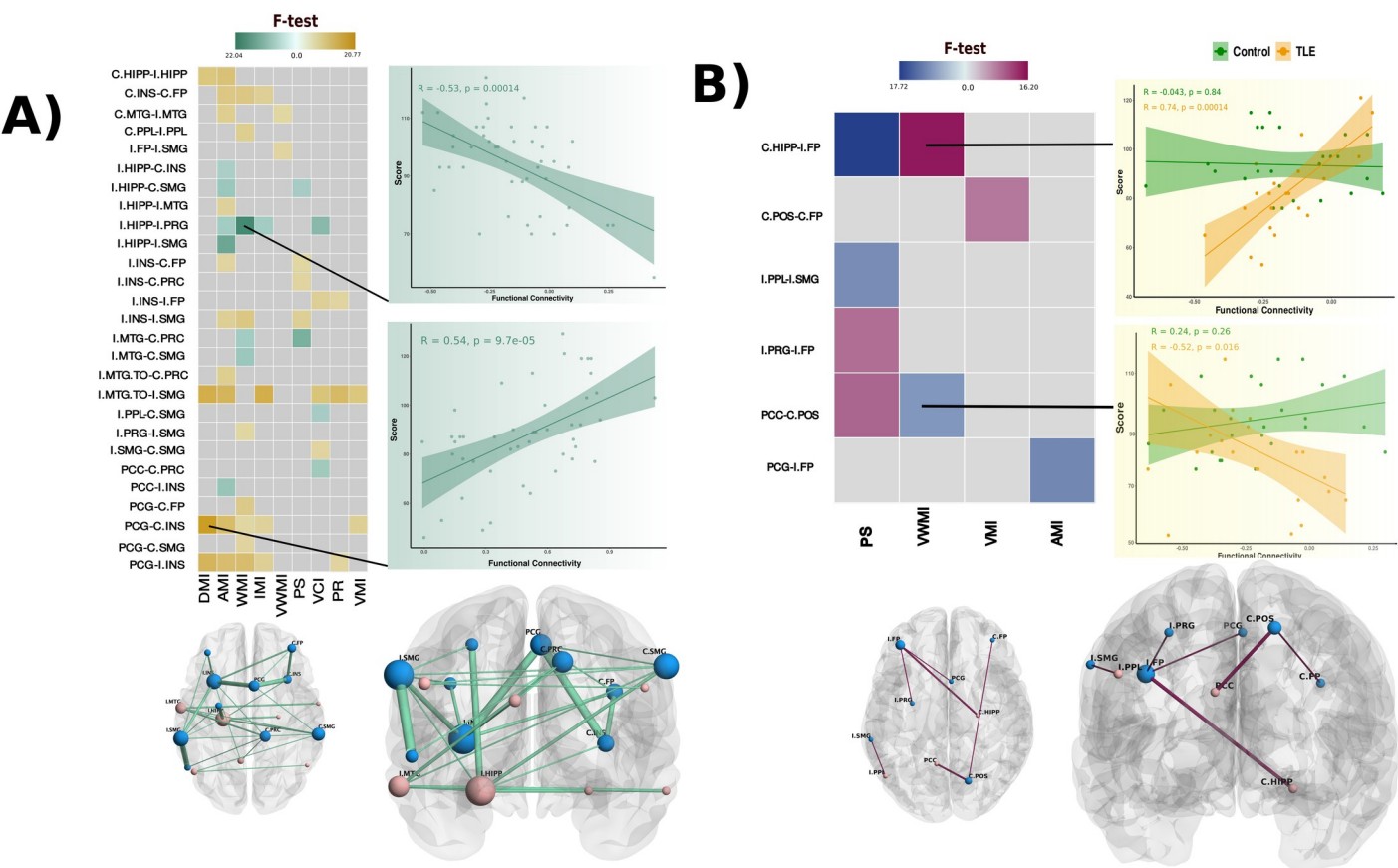

**Fig 4. Functional connectivity and cognitive scores.** A) Significant associations between FC and cognitive scores in all participants (GLM) and B) associations between FC and cognitive scores with significant group interaction (ANCOVA). In the matrices, colored cells depict significant F-test values (p < 0.05). Scatter plots on the right side are shown to illustrate the linear relationships or group interactions between node-node pairs and cognitive scores (all scatter plots are available in S4C and S4D Fig). Functional connectivity is expressed as Fisher's z. Pearson coefficients and p-values are shown color-coded for significant group-specific correlations. At the bottom, spatial representations of significant associations are depicted. Light-red colored spheres represent DMN nodes and blue colored spheres represent TPN nodes. Size of spheres (nodes) is proportional to the frequency of involvement of a given node in any significant association. The thickness of sticks (edges) is proportional to the frequency of involvement of a given node-node pair in any significant association. Results for GLM and ANCOVA models adjusted for clinical information are shown in S4E Fig.

group×digits was found (F(2,134) = 0.744, p = 0.47). We analyzed the mean reaction time of subjects along correct response trials. Mixed-effects ANOVA revealed significant effects of group (F(1,47) = 6.44, p < 0.05) and cognitive load (F(2,94) = 17.102, p < 0.0001) but no significant interaction between these factors (F(2,94) = 2.942, p = 0.058). Post-hoc comparisons revealed that in comparison to controls, TLE patients had significantly greater reaction times when they had to memorize either 2 digits (t(47) = -3.39, p < 0.01) or 4 digits (t(47) = -2.09, p < 0.05). Additionally, post-hoc tests revealed that only control subjects significantly increased their reaction time in the 4 (t(20) = -4.37, p < 0.001) and 6 digits (t(20) = -5.77, p < 0001) conditions in comparison with the 2 digits condition. Fig 5A illustrates these differences.

**Functional correlates of Sternberg memory task.** We fitted a GLM to find the significant brain regions involved in the retention phase of the WMem task. In both groups, we observed bilateral activation of frontal poles, supramarginal gyri, anterior insular cortices and middle frontal gyri (Fig 5B). Furthermore we detected activation in the Ipsilateral (left) precentral gyrus, and the Juxtapositional Lobule cortex/Paracingulate gyrus. When comparing between-

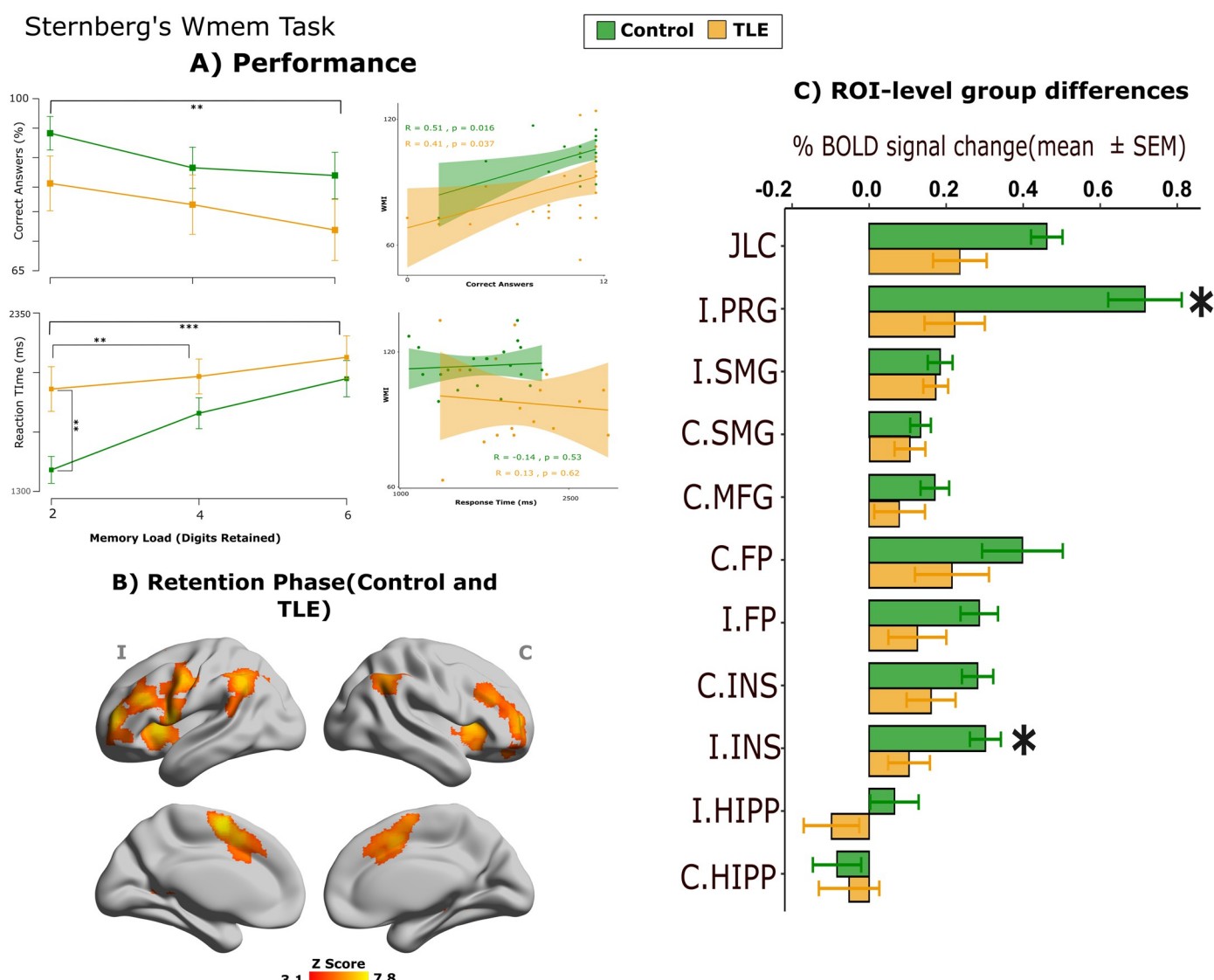

**Fig 5. Sternberg's working memory task.** A) Correct answers and reaction time (mean and standard error) as a function of memory load and its correlation with WMI derived from cognitive batteries. B) Group-level fMRI activation (controls and patients) during retention phase. Statistical map is shown referenced to the seizure onset hemisphere (I: ipsilateral; C: contralateral). C) Significant between-group voxelwise differences on fMRI activation during retention phase. Asterisks indicate $p_{FDR} < 0.05$.

group activity, planned contrasts revealed that TLE patients had less BOLD activity than controls in ipsilateral precentral gyrus. Contrarily, contralateral frontal operculum had significantly greater activity in TLE patients. In all cases, controls display greater BOLD signal change as compared to TLE patients.

**Association between BOLD activity during retention phase and cognitive scores.** Voxelwise analyses revealed that the correlation between cognitive indexes and BOLD activity within supramarginal gyri, intraparietal sulci, ipsilateral precentral gyrus and contralateral occipital cortex significantly differed between groups. Fig 6 illustrates correlations between mean BOLD percentage change during Sternberg's task and cognitive indexes that differed between patients and controls. Group interaction showed different directions of slopes, in all cases showing patients having significant correlations between the two variables.

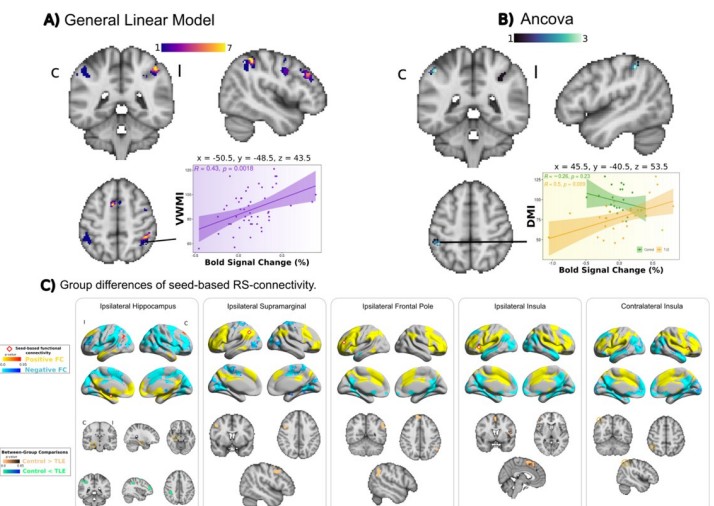

**Fig 6.** A: Correlations between BOLD activity and cognitive scores. A scatter plot is shown for illustrative purposes for visual working memory index. B: ANCOVAs showed positive correlations only in TLE patients, and not in controls for three different memory indices. Analysis was restricted to regions involved in Sternberg's task, and shown corrected for family-wise errors. Results of voxel-wise GLM and ANCOVA for the rest of cognitive scores are shown in S5 Fig. Color bars in A and B indicate the number of cognitive indexes that show significant correlations. C: Voxelwise analyses of seed-based RSFC for ROIs derived from the retention stage of Sternberg's memory task. Top row shows connectivity of the whole brain RSFC of seed (red-outline diamonds). Bottom row shows group-wise differences of the corresponding connectivity maps.

**Alterations in RS-FC in retention phase (ROI-analysis).** Finally, we hypothesized that TLE patients would display alterations of RSFC related to regions involved in the retention phase of Sternberg's memory task. To this end, we used seed-based connectivity analyses (SCA) and evaluated the correlation of the BOLD signal between the seed region and the rest of the brain. Brain areas found to be active in the retention phase of the WMem task (Fig 6) and both hippocampi served as seed regions. We found significant group-level differences in the connectivity of the following areas: both insular cortices, both hippocampi, ipsilateral frontal pole and ipsilateral supramarginal gyrus. Broadly, we observed that patients had decreased RSFC in salience network (as evidenced by insular SCA) and ipsilateral fronto-parietal network (as evidenced by ipsilateral supramarginal and ipsilateral frontal pole SCA). Large between-group differences were found in hippocampal SCA. In general, patients displayed decreased RSFC between C.HIPP and I.HIPP and greater RSFC between C.HIPP and Contralateral fronto-parietal network regions. All of these results were consistent and similar to those obtained through the network-based approach, showing abnormal RSFC in TPN, DMN and in anticorrelations between both systems.

## Discussion

Cognitive deficits are common in patients with TLE and affect several cognitive domains that involve brain regions within and beyond the temporal lobes. Here, we confirm that, as a group, TLE patients have cognitive deficits across domains, albeit with considerable inter-subject heterogeneity. Through analysis of resting-state fMRI we showed that cognitive deficits are associated with disorganized functional connectivity, with patients showing decreased connectivity within the default-mode and temporo-polar networks and, contrastingly, a less marked decoupling between DMN and TPN than what is typically seen in healthy individuals. Extensive cognitive testing allowed us to identify that the degree of specific region-to-region

connectivity is associated with neuropsychometric scores of visual memory, processing speed, and perceptual reasoning. Further testing of working memory within the imaging session showed that alterations of WMem are accompanied by reduced activity of temporo-parietal regions, which are also less tightly connected to remote brain areas. Our work provides evidence that the association between functional connectivity and cognitive scores is modulated by the presence of epilepsy and adds information related to the various forms of cognitive deficits seen in patients with TLE.

In previous studies with the same sample of participants, it has been observed that cognitive performance is not homogeneous across TLE patients, who show cognitive scores ranging from intact to severe generalized cognitive deficits [21, 31]. The range and type of cognitive disabilities seen in TLE patients have in turn been associated with specific patterns of morphological alterations [15] and structural connectivity [17, 48]. Therefore, it is believed that the distributed gray [13] and white matter damage [12] seen in TLE explain the heterogeneity of the type and severity of cognitive deficits in these patients through a disruption of the normal flow of information within the brain. Indeed, abnormalities of functional connectivity have been reported in TLE, providing evidence for abnormalities of network topography [49, 50] and alterations of the DMN, the attention network, and the reward/emotion network [29, 51–56]. Spatial patterns of RSFC abnormalities greatly coincide with structural damage [20, 57], and appear to be mediated by microstructural abnormalities of white matter [56]. Normally, brain regions within the TPN increase their activity during cognitive tasks, and are negatively correlated with components of the DMN [58]. Our results recapitulate the previously observed reduction of functional connectivity within each of the DMN and TPN. Moreover, TLE patients failed to show anticorrelations between DMN and TPN, while healthy participants displayed said typical association (Fig 3) [23, 59]. In agreement with previous reports, inter-hippocampal connectivity was decreased in TLE patients, as was the interaction between the ipsilateral hippocampus and temporal neocortex [60–62]. When accounting for demographic and clinical variables, the majority of findings were reproduced with a similar magnitude and direction (S3 Fig). A reduction in the effect was observed when adjusting for duration of illness and structural integrity of hippocampi (volume and presence of hippocampal sclerosis). Thus, these variables are an important source of variance that account for the observed FC alterations in TLE patients. This effect was particularly noticeable in edges that involved ipsilateral hippocampus and ipsilateral medial temporal lobe nodes. Interestingly, adjusting for these covariates also revealed alterations in contralateral hippocampus-TPN edges. These alterations, as well as decreases of DMN-TPN anticorrelations may account for the cognitive deficits observed in the TLE group.

We observed that FC within DMN or TPN edges is positively correlated with cognitive performance, while FC between DMN-TPN edges is rather negatively correlated (Fig 4A). In addition, we found several instances where TLE and healthy participants displayed different associations between region-to-region connectivity and cognitive performance (Fig 4). The majority of these between-group differences consisted of significant correlations seen in patients, but not in controls. It is tempting to associate increased functional connectivity with better performance, and there are examples advocating for that view in our results, and in the literature [63]. Nonetheless, the opposite can be observed in some connectivity edges. In our study we found that visual working memory was poorer in TLE patients with higher degree of connectivity between the posterior cingulate cortex and the contralateral parieto-occipital sulcus. Increased connectivity between these two nodes also had a negative effect on processing speed, an this negative effect was also seen for the increased connectivity between ipsilateral posterior parietal lobule and ipsilateral supramarginal gyrus, and ipsilateral precentral gyrus and ipsilateral frontal pole. The reconfiguration of functional networks suggest compensatory

mechanisms at play in patients to maintain cognitive abilities [57]. Similar modifications of brain networks related to WMem have also been reported in infants with periventricular leukomalacia [64].

There are discrepancies in our results as compared to previous reports. First, we do not see an increase of connectivity of the contralateral mesiotemporal networks, which has been shown to correlate with WMem performance [65]. Increased functional connectivity between ipsilateral mesial temporal structures and the ipsilateral posterior DMN have been reported to have an association with poorer verbal memory abilities [60, 65], while increased coupling between the ipsilateral hippocampus and contralateral posterior DMN shows a positive relation with improved verbal memory [66]. In our study, the connectivity of the ipsilateral hippocampus to other structures (contralateral hippocampus, insula, supramarginal gyrus, and ipsilateral middle temporal, precentral, and supramarginal gyri) was associated with auditory, working, and immediate memory indexes. Notably, delayed memory index was positively associated with the connectivity between both hippocampi. Conversely, increased connectivity between the ipsilateral hippocampus and ipsilateral precentral gyrus negatively correlated with auditory, working and immediate memory indexes, as well as verbal comprehension index. The contralateral left mesial temporal lobe and the ipsilateral medial prefrontal cortex has previously been shown to be positively associated with higher scores for non-verbal memory [60]. In our study, TLE patients (but not control subjects) showed a similar association (Fig 4B). While some of the previous studies studied left and right TLE patients separately, we pooled both patient groups and studied their brains according to ipsi- or contralateral to seizure onset. This, added to the known functional brain asymmetries behind many cognitive abilities (e.g., verbal) likely accounts for the differences between studies.

Despite having clear deficiencies in working memory when evaluated outside the scanner (Fig 2), TLE patients in our study showed only a tendency for reduced correct answers during Sternberg's task in the MRI scanner, but there was a clear difference between the two groups in their reaction times, with patients responding much later than controls, even in trials with minimal cognitive load (Fig 5). Correspondingly, the large majority of brain regions that form the normal fronto-parietal network involved in WMem [28] showed less activation in TLE patients than healthy participants, replicating and extending previous findings [30]. Moreover, the level of activation of fronto-parietal regions involved in WMem showed differential correlations with other immediate and delayed memory indices, with positive correlations seen in TLE patients, but not in healthy participants (Fig 6), suggesting compensatory mechanisms to maintain WMem. Finally, seed-based analyses of resting-state BOLD co-activation showed larger connectivity between the hippocampi and fronto-parietal regions in TLE patients, as compared to controls (Fig 6). This result is in line with the increased co-activation of these same regions seen during an N-back WMem task [18]. In said report, as in ours, increased co-activation of these brain regions was associated with poor WMem performance. Our work extends said observations by showing that TLE patients have decreased connectivity between ipsilateral insula and frontal regions, and between ipsilateral frontal pole and supramarginal gyrus. Thus, our results show the WMem network of TLE patients is altered at rest, is suboptimally engaged during memory retention, and is differentially modulated to subserve immediate and delayed memory performance.

There are some limitations in our study. First, we could not obtain histological confirmation of hippocampal sclerosis as the patients studied, unfortunately, did not undergo surgical treatment. Despite best efforts from the medical community, there are reduced opportunities for surgical treatment of TLE in low- and middle-income countries [67]. Also, in an effort to boost statistical power, we congregated TLE patients regardless of the hemisphere of seizure onset, and analyzed their data according to ipsilateral or contralateral to the epileptogenic

temporal lobe. This strategy makes it difficult to adequately interpret results related to cognitive or functional hemispheric specialization, such as verbal skills in our resting-state connectivity analyses. On the other hand, findings regarding the bilateral and symmetrical network involved in WMem are likely less affected by this limitation.

In conclusion, our results allowed us to link whole-brain connectivity with cognitive scores in multiple domains, showing abnormal patterns in TLE patients that tie together with cognitive deficits, as well as putative compensatory mechanisms. Targeting one specific cognitive feature, namely WMem, we were able to identify abnormalities in the activity of the underlying brain regions in TLE, but also how these associate to other brain regions and networks and affect several other cognitive abilities.

## Supporting information

**S1 Fig. Included subjects.**
(TIF)

**S2 Fig. Between-group comparison of neuropsychological scores.**
(TIF)

**S3 Fig. Functional connectivity adjusted for covariates.** p-value < 0.05, uncorrected.
(TIF)

**S4 Fig. C:** Scatterplots of GLM significant associations (n = 46). **D:** Scatterplots of ANCOVA significant associations (n = 46). **E:** Associations between Cognitive scores and Functional connectivity adjusted for covariates.
(ZIP)

**S5 Fig.** Voxel-wise results for GLM (A) and ANCOVA (B) for all cognitive scores. Interactive maps are available at https://github.com/alffajardo/TLE2023_fMRI.
(TIF)

## Acknowledgments

We thank the patients and control subjects for their willingness to participate, the medical specialists who helped us with their recruitment, and the clinical personnel at the National Laboratory for magnetic resonance imaging. We are grateful to Juan Ortíz-Retana and Leopoldo González-Santos for technical assistance. We thank the National Laboratory for scientific visualization (LAVIS) and staff, namely Luis Aguilar and Alejandro de León. We are grateful to the many people who have at some point participated in this project: Leticia Velázquez-Pérez, David Trejo, Héctor Barragán, Arturo Domínguez, Ildefonso Rodríguez-Leyva, Ana Luisa Velasco, Luis Octavio Jiménez, Daniel Atilano, Elizabeth González Olvera, Rafael Moreno, and Ana Elena Rosas.

## Author Contributions

**Conceptualization:** Alfonso Fajardo-Valdez, Vicente Camacho-Téllez, Raúl Rodríguez-Cruces, María Luisa García-Gomar, Erick Humberto Pasaye, Luis Concha.

**Data curation:** Alfonso Fajardo-Valdez, Vicente Camacho-Téllez, Raúl Rodríguez-Cruces, Erick Humberto Pasaye, Luis Concha.

**Formal analysis:** Alfonso Fajardo-Valdez, Vicente Camacho-Téllez, Luis Concha.

**Funding acquisition:** Luis Concha.

**Investigation:** Alfonso Fajardo-Valdez, Vicente Camacho-Téllez, Raúl Rodríguez-Cruces, Luis Concha.

**Methodology:** Alfonso Fajardo-Valdez, Vicente Camacho-Téllez, Raúl Rodríguez-Cruces, María Luisa García-Gomar, Erick Humberto Pasaye, Luis Concha.

**Project administration:** Luis Concha.

**Resources:** Erick Humberto Pasaye, Luis Concha.

**Software:** María Luisa García-Gomar, Luis Concha.

**Supervision:** Erick Humberto Pasaye, Luis Concha.

**Validation:** Alfonso Fajardo-Valdez, Luis Concha.

**Visualization:** Alfonso Fajardo-Valdez, Vicente Camacho-Téllez, Luis Concha.

**Writing – original draft:** Alfonso Fajardo-Valdez, Luis Concha.

**Writing – review & editing:** Vicente Camacho-Téllez, Raúl Rodríguez-Cruces, María Luisa García-Gomar, Erick Humberto Pasaye.

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
