## [Decision Letter · Decision Letter 0]

7 May 2023

PONE-D-23-04338Functional Correlates of Cognitive Performance and Working Memory in Temporal Lobe Epilepsy: Insights from Task-based and Resting-state fMRIPLOS ONE

Dear Dr. Concha,

Thank you for submitting your manuscript to PLOS ONE. After careful consideration, we feel that it has merit but does not fully meet PLOS ONE’s publication criteria as it currently stands. Therefore, we invite you to submit a revised version of the manuscript that addresses the points raised during the review process.

We look forward to receiving your revised manuscript.

Kind regards,

Florian Ph.S Fischmeister

Academic Editor

PLOS ONE

Journal Requirements:

"This work was supported by CONACYT (LC: 181508, 1782); and UNAM-DGAPA (LC: IB201712, IG200117, IN204720). RR-C and AF received fellowships from Conacyt (329866 and 478686). Imaging was 

performed at the National Laboratory for magnetic resonance imaging, which has received funding from CONACYT."

"This work was supported by CONACYT (LC: 181508, 1782); and UNAM-DGAPA (LC: IB201712, IG200117, IN204720). RR-C and AF received fellowships from Conacyt (329866 and 478686). Imaging was 

performed at the National Laboratory for magnetic resonance imaging, which has received funding from CONACYT."

5. We note that Figures 3, 4, 5 and 6 in your submission contain copyrighted images. All PLOS content is published under the Creative Commons Attribution License (CC BY 4.0), which means that the manuscript, images, and Supporting Information files will be freely available online, and any third party is permitted to access, download, copy, distribute, and use these materials in any way, even commercially, with proper attribution. For more information, see our copyright guidelines: http://journals.plos.org/plosone/s/licenses-and-copyright.

a. You may seek permission from the original copyright holder of Figures 3, 4, 5 and 6 to publish the content specifically under the CC BY 4.0 license. 

Reviewers' comments:

Reviewer's Responses to Questions

**Comments to the Author**

1. Is the manuscript technically sound, and do the data support the conclusions?

Reviewer #1: Yes

Reviewer #2: Yes

2. Has the statistical analysis been performed appropriately and rigorously? 

Reviewer #1: Yes

Reviewer #2: Yes

3. Have the authors made all data underlying the findings in their manuscript fully available?

Reviewer #1: No

Reviewer #2: Yes

4. Is the manuscript presented in an intelligible fashion and written in standard English?

Reviewer #1: Yes

Reviewer #2: Yes

5. Review Comments to the Author

Reviewer #1: In this manuscript, Fajardo-Valdez, Camacho-Téllez et al. report interesting findings related to fMRI-derived functional connectivity in temporal lobe epilepsy patients during wakeful resting, and patterns of BOLD activation while performing a working memory task. The paper reports behavioural data (neuropsychological measures, working memory paradigm performance) and functional connectivity in both resting-state (DMN) and neurocognitive networks (TPN), and evaluates the (anti)correlation between them. The authors observe that the TLE patients exhibit decreased functional connectivity in the default mode and temporo-polar networks during rest, and decreased BOLD response in temporo-parietal regions during the working memory task. In my opinion, the findings constitute a valuable contribution to the evolving understanding of the consistently reported altered functional connectivity, in concurrence with neurocognitive deficits, observed in focal epilepsy patients. The manuscript is well-written and communicates well. Pending that the authors adequately address the remarks listed below, I can recommend the manuscript for publication.

Major remarks:

- The authors state that the sample is further described in a previous publication (Rodriguez-Cruces et al., 2020); however, it appears that the size of the sample used in the referenced paper is not identical to what is reported in the current manuscript. I ask the authors to clarify this issue. While I was able to find clinical information about a part (?) of the sample in a previous publication (Rodriguez-Cruces et al,, 2018), I strongly recommend that these data are reported for the current sample in the current manuscript (see the next remark for context). If the clinical data are already included in the published dataset at OpenNeuro - which I was unable to access (see minor comment below) - the authors may disregard this remark.

- Following the previous remark, I would strongly encourage the authors to provide more clinical information pertaining to the study participants, as correlations between such variables and functional connectivity have consistently been reported. In my opinion, the manuscript could be further improved by investigating if the reported effects (e.g., DMN-TPN anticorrelation) are mediated by clinical factors. I ask the authors to, at the very least, comment on if including such variables as confounders in the analyses would be beneficial, and possibly conduct the analyses. Previously reported variables include (but are not limited to):

* Time with epilepsy/time since first seizure. Some studies indicate that epilepsy duration is associated with various functional connectivity-related characteristics (e.g., van Dellen et al., 2019, DOI: 10.1371/journal.pone.0008081).

* Medication. Was the TLE group homogeneous in terms of medication? Would it be possible to create sub-groups based on medications (e.g., Haneef et al., 2015, DOI: 10.1089/brain.2014.0304)?

* Surgery. The authors state that “the majority” of patients had not undergone surgery. Surgery has been demonstrated to alter functional connectivity (e.g., for a review, see Johnson et al., 2022, DOI: 10.1097/WCO.0000000000001008).

- I ask the authors to more clearly describe and justify the data rejection procedures employed in the manuscript. To my understanding, while a relatively large dataset is introduced (TLE = 40; HC = 36), it appears that considerably reduced datasets were used in the resting-state fMRI analysis (TLE = 21; HC = 25), and the task-related fMRI analysis (TLE = 33; HC = 33). The stated reason for the reduction is poor fMRI data quality following visual inspection. Based on the information available, I feel obliged to characterise this approach as questionable from a strict methodological standpoint. Considering that almost half of the TLE patients’ data were rejected for the resting-state analysis, I would strongly encourage the authors to carefully detail the rejection procedures. Ideally, objective metrics should be used to facilitate reproducibility. Please clarify this point carefully.

- In a remark related to the previous question about the data rejection procedures, I wonder how many participants were included in the analysis of the neuropsychological test scores? If my understanding is correct, the observed (significant) differences in performance on the neuropsychological tests were based on group comparisons using the full dataset (TLE = 40 vs. HC = 36). If this is the case, I would argue that the analysis could be considered misleading, due to the fact that the analysed sample does not match the samples used for the resting-state and task-related fMRI analyses. While I appreciate that the correlation analyses were conducted using only data from subjects whose fMRI data were retained, it is, in my opinion, problematic that the manuscript draws conclusions regarding the TLE patients’ neuropsychological performance based the whole TLE group when only half of the patients are considered in the analyses that are presented as the manuscript’s main findings. If my assumptions are indeed true, I would suggest that the authors compare neuropsychological test performance between TLE patients and healthy controls in the sub-groups, i.e., subjects with retained resting-state data, and subjects with retained task-related data.

- I would also ask the authors to elaborate their methodological choice of flipping the x axis of right TLE patients. E.g., does this choice make the implicit assumption that all activations are bilaterally mirrored? Please justify why it is better methodologically to use the epileptic focus as a reference rather than anatomical structures. Has its applicability been documented anywhere (reference?)? I appreciate that the issue is mentioned in the discussion (towards the end of page 14), but I would encourage the authors to detail assumptions, implications, and limitations associated with this choice. Also, was the flipping done only for the resting-state data? Was the flipping reversed when compared directly with the activation ROIs during the Sternberg task (towards the end of page 7)?

- In Results, “3.3. Associations between functional connectivity and cognitive abilities”, the authors state that “TLE patients showed several significant correlations between FC and cognitive abilities, while healthy subjects did not” while Figure 4 clearly shows several significant correlations for both groups, e.g., in working memory. Please clarify.

Minor remarks:

- A text piece is repeated in Methods. One states that 45 subjects were included for fMRI analysis while the other states 46 subjects.

- While not a strict requirement, I would encourage the authors to use more precise terminology with regards to the neuropsychological data in the manuscript. E.g., neuropsychological test data represents a performance-based cognitive index, not a direct measure of cognitive abilities.

- On the behavioural level, did you observe correlations between any of the neuropsychological working memory indices (e.g., WMI) and performance on the Sternberg task? If not, are they measuring the same cognitive operation?

- For enhanced readability, I recommend restricting the use of the terms “activity” and “activation” to the context of BOLD activation. In the context of functional connectivity, the terms are imprecise, as connectivity is, per definition, the interaction between two spatially separated “activations”.

- I was not able to access the OpenNeuro dataset. While this is not crucial for the review, it is possible that the answer to some of my remarks lie therein.

- While the figures for the most part are informative, the figure captions could benefit from some light revision. Make sure all relevant information is present.

Reviewer #2: Thank you for giving me the opportunity to review this interesting manuscript.

Concha et al. report an fMRI study (resting-state and task-based using a modified Sternberg task) to evaluate anomalies in the working memory (WMem) network in 40 people with TLE and 36 healthy controls, and how WMem network impairments affect overall cognitive performance.

Overall, the results confirm previous findings of reduced connectivity within DMN and/or TPN, an impaired anticorrelation of DMN and TPN and reduced inter-hippocampal connectivity in TLE.

There are some discrepancies to previous reports regarding correlations of BOLD activity/rs connectivity with cognitive scores, e.g. the finding that contralateral hippocampus connectivity to ipsilateral MTG and contralateral PPL correlated negatively with VWMI and processing speed. As the authors state, these differences may in part be attributed to differences in methodological aspects, particularly the analysis of data according to ipsilateral or contralateral to the epileptogenic temporal lobe.

The paper is well written, and the topic itself is of great interest, as cognitive deficits (e.g. memory and language deficits) are frequent concerns in people with TLE with significant detrimental impacts on quality of life. The manuscript is technically sound, the data support the conclusions, methodology and statistical analyses are clearly described and performed appropriately. Data underlying the findings are made available by the authors.

However, with respect to the existing literature on the topic, research questions and hypotheses should be clarified. In this regard, the Discussion would benefit from rewriting, should clearly highlight the results that go beyond previous findings, and include an elaboration how the current results may impact the future care of people with TLE.

Minor points:

• Typo page 3, section 2.2 „Weschler’s Memory Scale“

• Page 6, section 2.8: „..ANCOVA models can be described with by the following linear model equation“ – suggest to rephrase to „..described using the…“

• Page 7, section 2.10: „Mean BOLD percentage change during the retention phase relative to basal activity was obtained per ROI and was analyzed and compared between groups“

Could the authors elaborate on the baseline condition that was used to assess relative signal change?

• Page 10, 3.3:„TLE patients showed several significant correlations between FC and cognitive abilities, while healthy subjects did not. These associations involved both hicompampi, middle temporal lobe regions, insular cortices and Ipsilateral precentral gyrus“

Could the authors elaborate more on the correlations that were observed in TLE (and which are displayed in Figure 4)? The colour coding in Figure 4 seems a bit confusing: on the upper left, red and blue indicate DMN and TPN, respectively, while in the scatterplots, red and blue indicate TLE vs. controls? In the scatterplots, it is not intuitive which correlations/anticorrelations involve DMN or TPN? Also, the authors state that no correlations were observed in controls, however, some of the scatterplots (e.g. working memory) appear to indicate a correlation in controls rather than TLE?

6. PLOS authors have the option to publish the peer review history of their article (what does this mean?). If published, this will include your full peer review and any attached files.

Reviewer #1: No

Reviewer #2: No

---

## [Author Response · Author response to Decision Letter 0]

12 Oct 2023

Functional Correlates of cognitive performance and working memory in temporal lobe epilepsy: Insights from task-based and resting-state fMRI.

We are grateful for the opportunity to submit a revised version of our work, and thank the two Reviewers for their careful reviews and thoughtful comments. Through this revision we were able to clarify important points and have taken the opportunity to add supplementary material, as well as several interactive figures.

Response to Reviewers

Reviewer #1 

Summary

In this manuscript, Fajardo-Valdez, Camacho-Téllez et al. report interesting findings related to fMRI-derived functional connectivity in temporal lobe epilepsy patients during wakeful resting, and patterns of BOLD activation while performing a working memory task. The paper reports behavioural data (neuropsychological measures, working memory paradigm performance) and functional connectivity in both resting-state (DMN) and neurocognitive networks (TPN), and evaluates the (anti)correlation between them. The authors observe that the TLE patients exhibit decreased functional connectivity in the default mode and temporo-polar networks during rest, and decreased BOLD response in temporo-parietal regions during the working memory task. In my opinion, the findings constitute a valuable contribution to the evolving understanding of the consistently reported altered functional connectivity, in concurrence with neurocognitive deficits, observed in focal epilepsy patients. The manuscript is well-written and communicates well. Pending that the authors adequately address the remarks listed below, I can recommend the manuscript for publication.

We thank the Reviewer for the positive and constructive comments.

Major remarks:

- The authors state that the sample is further described in a previous publication (Rodriguez-Cruces et al., 2020); however, it appears that the size of the sample used in the referenced paper is not identical to what is reported in the current manuscript. 

1. I ask the authors to clarify this issue. While I was able to find clinical information about a part (?) of the sample in a previous publication (Rodriguez-Cruces et al,, 2018), 

2. I strongly recommend that these data are reported for the current sample in the current manuscript (see the next remark for context). If the clinical data are already included in the published dataset at OpenNeuro - which I was unable to access (see minor comment below) - the authors may disregard this remark.

We recognize the lack of clarity in the original description of our sample size. Detailed information has been added to the article, and for the sake of simplicity we summarize here.

Our original sample consisted of more than 80 consecutive participants. From these, we selected a total 66 subjects that met the inclusion criteria. Of the 66 participants, 31 were TLE patients and 35 were control subjects. The different sub-analyses had specific data curation requirements, which caused sample sizes to differ between the task-fMRI and RS-FMRI analyses. We have now clearly indicated the sample size for each sub-analysis and provide a supplementary figure (S1 Fig) that details the sub-analyses in which each participant was included. Moreover, this is reflected in the participants.tsv file in the openly available data set.

The reasons for the mismatch in sample sizes between sub-analyses include:

Incomplete neuropsychological evaluations (the most common cause, which occurred in 14 participants).

Insufficient usable RS-fMRI volumes (less than 4 minutes) after preprocessing and motion assessment based on FD > 0.25 (based on Power et al., Neuroimage 2013; PMC3849338. See below for more details). Occurred in five participants.

Severe signal and encoding artifacts or signal dropout in temporal or frontal lobes (occurred only in two subjects).

- Following the previous remark, 

3. I would strongly encourage the authors to provide more clinical information pertaining to the study participants, as correlations between such variables and functional connectivity have consistently been reported. In my opinion, the manuscript could be further improved by investigating if the reported effects (e.g., DMN-TPN anticorrelation) are mediated by clinical factors. I ask the authors to, at the very least, comment on if including such variables as confounders in the analyses would be beneficial, and possibly conduct the analyses. Previously reported variables include (but are not limited to):

 - Time with epilepsy/time since first seizure. Some studies indicate that epilepsy duration is associated with various functional connectivity-related characteristics (e.g., van Dellen et al., 2019, DOI: 10.1371/journal.pone.0008081).

 - Medication. Was the TLE group homogeneous in terms of medication? Would it be possible to create sub-groups based on medications (e.g., Haneef et al., 2015, DOI: 10.1089/brain.2014.0304)?

 - Surgery. The authors state that “the majority” of patients had not undergone surgery. Surgery has been demonstrated to alter functional connectivity (e.g., for a review, see Johnson et al., 2022, DOI: 10.1097/WCO.0000000000001008).

Thank you for these valuable suggestions. We re-analyzed our data including clinical information as covariates, with a particular interest in age at diagnosis and hippocampal volume. Overall, edge-wise results were similar at the uncorrected p value (S3 Fig), Notably, nine edges consistently showed between-group differences in all the explored models, involving both hippocampi, middle temporal gyri, and contralateral insula and supramarginal gyrus. However, these models did not show statistically-significant differences after correction with FDR and NBS. This is now included in the section Network analysis.

None of the patients studied were on benzodiazepines, barbiturates, or topiramate, all known to affect cognitive functions (this was an important exclusion criteria that was not mentioned in the first version of our paper and is now included in Methods). Despite this, medication regimes were heterogeneous amongst the patient sample included. There was also heterogeneity in terms of the number of antiseizure medications used at the time of scanning. For these reasons, we found it difficult to create reliable sub-groups for further analysis. 

Our patients were all recruited from out-patient clinics, as mentioned in Methods. Sadly, while many patients constitute ideal candidates for epilepsy surgery, this treatment option was not readily available for them. We had used the wording “the majority of patients studied, unfortunately, did not undergo surgical treatment”. We confirmed that none of the patients underwent surgery, and have corrected this sentence accordingly. We apologize for this mistake.

4. I ask the authors to more clearly describe and justify the data rejection procedures employed in the manuscript. To my understanding, while a relatively large dataset is introduced (TLE = 40; HC = 36), it appears that considerably reduced datasets were used in the resting-state fMRI analysis (TLE = 21; HC = 25), and the task-related fMRI analysis (TLE = 33; HC = 33). The stated reason for the reduction is poor fMRI data quality following visual inspection. Based on the information available, I feel obliged to characterise this approach as questionable from a strict methodological standpoint. Considering that almost half of the TLE patients’ data were rejected for the resting-state analysis, I would strongly encourage the authors to carefully detail the rejection procedures. Ideally, objective metrics should be used to facilitate reproducibility. Please clarify this point carefully.

As mentioned in the response to the first point, the main reason to exclude data from analyses searching for correlations between cognitive scores and (task and RS) fMRI features was an incomplete neuropsychological evaluation, rather than data rejection. Reasons to exclude data after pre-processing fMRI time series are included in section Resting-State fMRI preprocessing, which essentially follow the recommendations by Power 2012 and Satterthwaite 2013, as implemented in the CPAC pipeline. 

Power, Jonathan D., et al. "Spurious but systematic correlations in functional connectivity MRI networks arise from subject motion." Neuroimage 59.3 (2012): 2142-2154.

Satterthwaite, Theodore D., et al. "An improved framework for confound regression and filtering for control of motion artifact in the preprocessing of resting-state functional connectivity data." Neuroimage 64 (2013): 240-256.

5. In a remark related to the previous question about the data rejection procedures, I wonder how many participants were included in the analysis of the neuropsychological test scores? 

The main between-group comparison of neuropsychological test scores included 53 participants (28 patients 25 controls). We now include analyses of neuropsychological test scores for each sub-sample, and the number of participants is stated in each section, accordingly (S2 Fig)

6. If my understanding is correct, the observed (significant) differences in performance on the neuropsychological tests were based on group comparisons using the full dataset (TLE = 40 vs. HC = 36). If this is the case, I would argue that the analysis could be considered misleading, due to the fact that the analysed sample does not match the samples used for the resting-state and task-related fMRI analyses. While I appreciate that the correlation analyses were conducted using only data from subjects whose fMRI data were retained, it is, in my opinion, problematic that the manuscript draws conclusions regarding the TLE patients’ neuropsychological performance based the whole TLE group when only half of the patients are considered in the analyses that are presented as the manuscript’s main findings. If my assumptions are indeed true, 

7. I would suggest that the authors compare neuropsychological test performance between TLE patients and healthy controls in the sub-groups, i.e., subjects with retained resting-state data, and subjects with retained task-related data.

Points 6 and 7 raised by the Reviewer are correct and we appreciate the comments. As mentioned in the response to point 5, we performed additional independent analyses of cognitive scores including only (i) the sample of patients who underwent resting-state fMRI, or (ii) those with task-fMRI data. Group differences are very similar in these sub-samples. We now include these additional results and provide an additional S2 Fig.

8. I would also ask the authors to elaborate their methodological choice of flipping the x axis of right TLE patients. E.g., does this choice make the implicit assumption that all activations are bilaterally mirrored? Please justify why it is better methodologically to use the epileptic focus as a reference rather than anatomical structures. Has its applicability been documented anywhere (reference?)? I appreciate that the issue is mentioned in the discussion (towards the end of page 14), but I would encourage the authors to detail assumptions, implications, and limitations associated with this choice. Also, was the flipping done only for the resting-state data? Was the flipping reversed when compared directly with the activation ROIs during the Sternberg task (towards the end of page 7)?

Flipping the x axis of patients with TLE is a common approach to increase statistical power (e.g. Bernhardt et al, Ann Neurol 2016, Li et al., NeuroImage:Clin 2019, Morgan et al., J Neurosurg 2019). Moreover, referring to hemispheres as ipsilateral and contralateral to seizure focus simplifies data analysis and interpretation of results. As the Reviewer notes, however, this approach is not without caveats, and becomes relevant when studying cognitive abilities, which are known to show important inter-hemispheric asymmetries. Unfortunately, our sample size is not large enough to perform individual analyses of left/right TLE and left/right hemispheres. We acknowledge this limitation in the Discussion.

We have further clarified that flipping of the data was done in the resting-state fMRI data, as well as in the activation ROIs during the Sternberg task.

Bernhardt, Boris C., et al. "The spectrum of structural and functional imaging abnormalities in temporal lobe epilepsy." Annals of neurology 80.1 (2016): 142-153.

Li, Wei, et al. "Different patterns of white matter changes after successful surgery of mesial temporal lobe epilepsy." NeuroImage: Clinical 21 (2019): 101631.

Morgan, Victoria L., et al. "Characterization of postsurgical functional connectivity changes in temporal lobe epilepsy." Journal of neurosurgery 133.2 (2019): 392-402.

9. In Results, “3.3. Associations between functional connectivity and cognitive abilities”, the authors state that “TLE patients showed several significant correlations between FC and cognitive abilities, while healthy subjects did not” while Figure 4 clearly shows several significant correlations for both groups, e.g., in working memory. Please clarify.

We have modified the paragraph that discusses these findings.

Minor remarks:

- A text piece is repeated in Methods. One states that 45 subjects were included for fMRI analysis while the other states 46 subjects.

We have corrected this, which is in line to the overall clarification of our sample sizes.

- While not a strict requirement, I would encourage the authors to use more precise terminology with regards to the neuropsychological data in the manuscript. E.g., neuropsychological test data represents a performance-based cognitive index, not a direct measure of cognitive abilities.

Several instances of "cognitive abilities" were changed to "cognitive scores".

- On the behavioural level, did you observe correlations between any of the neuropsychological working memory indices (e.g., WMI) and performance on the Sternberg task? If not, are they measuring the same cognitive operation?

Thank you for this suggestion. WMI correlated with the number of correct responses in Sternberg's task (r=0.47, p=0.0008). This is now included in the Results.

- For enhanced readability, I recommend restricting the use of the terms “activity” and “activation” to the context of BOLD activation. In the context of functional connectivity, the terms are imprecise, as connectivity is, per definition, the interaction between two spatially separated “activations”.

We appreciate the suggestion and have made several changes accordingly, reserving the word "activity" to refer to the BOLD response within a single brain area, and "co-activity" to refer to remote brain regions having significantly correlated fluctuations of BOLD signal.

- I was not able to access the OpenNeuro dataset. While this is not crucial for the review, it is possible that the answer to some of my remarks lie therein.

We apologize for not providing a link to a version of the data set available to Reviewers. The data set is now publicly available. We have taken this opportunity to extend the information available in the participants.tsv file in the repository, which should clarify questions regarding the sample size in the sub-analyses.

- While the figures for the most part are informative, the figure captions could benefit from some light revision. Make sure all relevant information is present.

Thank you for your suggestions. The figure captions have been revised.

Reviewer #2

Thank you for giving me the opportunity to review this interesting manuscript.

Concha et al. report an fMRI study (resting-state and task-based using a modified Sternberg task) to evaluate anomalies in the working memory (WMem) network in 40 people with TLE and 36 healthy controls, and how WMem network impairments affect overall cognitive performance.

Overall, the results confirm previous findings of reduced connectivity within DMN and/or TPN, an impaired anticorrelation of DMN and TPN and reduced inter-hippocampal connectivity in TLE.

There are some discrepancies to previous reports regarding correlations of BOLD activity/rs connectivity with cognitive scores, e.g. the finding that contralateral hippocampus connectivity to ipsilateral MTG and contralateral PPL correlated negatively with VWMI and processing speed. As the authors state, these differences may in part be attributed to differences in methodological aspects, particularly the analysis of data according to ipsilateral or contralateral to the epileptogenic temporal lobe.

The paper is well written, and the topic itself is of great interest, as cognitive deficits (e.g. memory and language deficits) are frequent concerns in people with TLE with significant detrimental impacts on quality of life. The manuscript is technically sound, the data support the conclusions, methodology and statistical analyses are clearly described and performed appropriately. Data underlying the findings are made available by the authors.

However, with respect to the existing literature on the topic, research questions and hypotheses should be clarified. In this regard, the Discussion would benefit from rewriting, should clearly highlight the results that go beyond previous findings, and include an elaboration how the current results may impact the future care of people with TLE.

We appreciate the positive feedback. The objectives of our study are outlined in the last paragraph of the Introduction. 

We have further clarified in the first paragraph of the Discussion how our work provides evidence for the modulation of cognitive scores due to the presence of epilepsy, and how this furthers our understanding of the heterogeneity of cognitive abilities seen in TLE patients. The rest of our Discussion highlights the important differences seen between our work and previous findings.

---

## [Decision Letter · Decision Letter 1]

5 Nov 2023

PONE-D-23-04338R1Functional Correlates of Cognitive Performance and Working Memory in Temporal Lobe Epilepsy: Insights from Task-based and Resting-state fMRIPLOS ONE

Dear Dr. Concha, Thank you for submitting your manuscript to PLOS ONE. After careful consideration, we feel that it has merit but does not fully meet PLOS ONE’s publication criteria as it currently stands. Therefore, we invite you to submit a revised version of the manuscript that addresses the points raised during the review process.

We look forward to receiving your revised manuscript.

Kind regards,

Florian Ph.S Fischmeister

Academic Editor

PLOS ONE

Journal Requirements:

Additional Editor Comments: 

Therefore, we invite you to submit a revised version of the manuscript that addresses the two minor points raised by reviewer 2 concering the additional analyses that should be elaborated in the discussion and a minor issue with missing statistical results for S2; c.f. below.

Reviewers' comments:

Reviewer's Responses to Questions

**Comments to the Author**

1. If the authors have adequately addressed your comments raised in a previous round of review and you feel that this manuscript is now acceptable for publication, you may indicate that here to bypass the “Comments to the Author” section, enter your conflict of interest statement in the “Confidential to Editor” section, and submit your "Accept" recommendation.

Reviewer #1: (No Response)

Reviewer #2: (No Response)

2. Is the manuscript technically sound, and do the data support the conclusions?

Reviewer #1: Yes

Reviewer #2: Yes

3. Has the statistical analysis been performed appropriately and rigorously? 

Reviewer #1: Yes

Reviewer #2: Yes

4. Have the authors made all data underlying the findings in their manuscript fully available?

Reviewer #1: Yes

Reviewer #2: Yes

5. Is the manuscript presented in an intelligible fashion and written in standard English?

Reviewer #1: Yes

Reviewer #2: Yes

6. Review Comments to the Author

Reviewer #1: I thank the authors for their comprehensive work addressing my remarks to the initial manuscript. I think the analyses considering clinical covariates were an interesting addition to the manuscript. While understandable, it was unfortunate that the heterogeneity with respect to medication prevented any subgroup analyses. Thank you for verifying that none of the patients had undergone surgery. If not very inconvenient for the authors, I would suggest that the medication and epilepsy onset variables are added to the participants.tsv file of the dataset.

Furthermore, I appreciate the authors’ efforts to clarify the dataset and which participants were included in which analyses. The revised manuscript promotes transparency on this issue in much greater extent than the original manuscript. In particular, the supplementary figure S1 was effective in this regard. Especially important was the explicit mentioning that data for the resting-state fMRI analyses were rejected mostly due to the lack of neuropsychological data, and not due to poor fMRI data quality.

I consider all my remarks adequately addressed, also those not explicitly mentioned above. I can recommend the manuscript for publication in its current state. Congratulations to the authors on the work.

Reviewer #2: Thank you for the opportunity to review this manuscript again.

I believe the concerns raised by both reviewers were addressed accordingly, and I only have two minor remarks:

1.) The additional analyses investigating whether clinical factors like age at onset of seizures, age at time of study, birth sex and presence of HS impact the observed effects should be elaborated on in the Discussion.

2.) S2 Fig: The authors state that S2 Fig shows similar/ nearly identical results to Figure 2, however, no statistical results for S2 Fig are presented in the Figure or text/caption.

7. PLOS authors have the option to publish the peer review history of their article (what does this mean?). If published, this will include your full peer review and any attached files.

Reviewer #1: **Yes: **Christoffer Hatlestad-Hall

Reviewer #2: No

---

## [Author Response · Author response to Decision Letter 1]

10 Nov 2023

Functional Correlates of cognitive performance and working memory in temporal lobe epilepsy: Insights from task-based and resting-state fMRI.

Response to Reviewers

We thank the two Reviewers for their overall positive feedback and appreciation of our work.

Reviewer #1 suggested the additon of information to the public data set, namely medication used by patients, and age of epilepsy onset. The new version of the data set now includes this information.

Reviewer #2 provided us with the following suggestions, which we have addressed in this version:

1.) The additional analyses investigating whether clinical factors like age at onset of seizures, age at time of study, birth sex and presence of HS impact the observed effects should be elaborated on in the Discussion.

We have included this in our Discussion (page 23), reproduced here:

When accounting for demographic and clinical variables, the majority of findings were reproduced with a similar magnitude and direction (S3 fig). A reduction in the effect was observed when adjusting for duration of illness and structural integrity of hippocampi (volume and presence of hippocampal sclerosis). Thus, these variables are an important source of variance that account for the observed  FC alterations in TLE patients. This effect was particularly noticeable in edges that involved ipsilateral hippocampus and ipsilateral medial temporal lobe nodes. Interestingly, adjusting for these covariates also revealed alterations in contralateral hippocampus-TPN edges. These alterations, as well as decreases of DMN-TPN anticorrelations may account for the cognitive deficits observed in the TLE group. 

2.) S2 Fig: The authors state that S2 Fig shows similar/ nearly identical results to Figure 2, however, no statistical results for S2 Fig are presented in the Figure or text/caption.

Statistical results were added to S2 Figure, as suggested.

---

## [Editor Report · Decision Letter 2]

16 Nov 2023

Functional Correlates of Cognitive Performance and Working Memory in Temporal Lobe Epilepsy: Insights from Task-based and Resting-state fMRI

PONE-D-23-04338R2

Dear Dr. Concha,

We’re pleased to inform you that your manuscript has been judged scientifically suitable for publication and will be formally accepted for publication once it meets all outstanding technical requirements.

Kind regards,

Florian Ph.S Fischmeister

Academic Editor

PLOS ONE

---

## [Editor Report · Acceptance letter]

20 Nov 2023

PONE-D-23-04338R2 

Functional Correlates of cognitive performance and working memory in temporal lobe epilepsy: Insights from task-based and resting-state fMRI. 

Dear Dr. Concha:

I'm pleased to inform you that your manuscript has been deemed suitable for publication in PLOS ONE. Congratulations! Your manuscript is now with our production department. 

Kind regards, 

on behalf of

Mag. Dr. Florian Ph.S Fischmeister 

Academic Editor

PLOS ONE